# CALM: CRITIC AUTOMATION WITH LARGE LANGUAGE MODELS

## ABSTRACT

Understanding the world through models is a fundamental goal of scientific research. While large language model (LLM) based approaches show promise in automating scientific discovery, they often overlook the importance of *criticizing* scientific models. Criticizing models deepens scientific understanding and drives the development of more accurate models. Moreover, criticism can improve the reliability of LLM-based scientist systems by acting as a safeguard against hallucinations. Automating model criticism is difficult because it traditionally requires a human expert to define how to compare a model with data and evaluate if the discrepancies are significant–both rely heavily on understanding the modeling assumptions and domain. Although LLM-based critic approaches are appealing, they introduce new challenges: LLMs might hallucinate the critiques themselves. Motivated by this, we introduce CALM (Critic Automation with Language Models). CALM uses LLMs to generate summary statistics that highlight discrepancies between model predictions and data, and applies hypothesis tests to evaluate their significance. We can view CALM as a verifier that validates models and critiques by embedding them in a hypothesis testing framework. In experiments, we evaluate CALM across key quantitative and qualitative dimensions. In settings where we synthesize discrepancies between models and datasets, CALM reliably generates correct critiques without hallucinating incorrect ones. We show that both human and LLM judges consistently prefer CALM's critiques over alternative approaches in terms of transparency and actionability. Finally, we show that CALM's critiques enable an LLM scientist to improve upon human-designed models on real-world datasets.

## 1 INTRODUCTION

A longstanding goal of artificial intelligence research is to automate the discovery of scientific models (Langley et al., 1987; Waltz & Buchanan, 2009). The rapid development of LLMs with remarkable reasoning capabilities and general knowledge has created exciting new opportunities within this domain. Recent work has shown that LLM based scientific agents can propose novel research ideas (Si et al., 2024), discover scientific models (Li et al., 2024), and implement experiments (Lu et al., 2024; Huang et al., 2024). These results highlight the promise of using LLMs to automate many important aspects of scientific discovery. However, they overlook the crucial role that *model criticism* plays in driving scientific progress. Understanding the limitations of existing models deepens our understanding and often motivates new models. Furthermore, automated methods for criticism can improve the reliability of LLM-based scientific discovery systems, as LLMs are prone to systematic hallucinations (Lu et al., 2024; Xu et al., 2024) that could undermine the broader goal of automating scientific discovery.

Model criticism is hard to automate because it is inherently dependent on the model and problem domain. In particular, it involves (1) determining which aspects to compare between the model and data and (2) evaluating the significance of any differences. Each of these tasks typically requires substantial human expertise (Gelman & Shalizi, 2012). While leveraging LLMs is an initially appealing approach to automation, it introduces new challenges: LLMs might also hallucinate the critiques themselves, undermining the effectiveness of automated model criticism.

Motivated by these challenges, we introduce **CALM** (Critic Automation with Language Models), which integrates LLMs within a principled model criticism framework. Specifically, given a proposed scientific model and dataset metadata, CALM uses an LLM to generate summary statistics that capture properties of the data that might violate the modeling assumptions. Importantly, these summary statistics are tailored to the model and dataset. CALM implements these summary statistics as `Python` functions, which can be easily executed and inspected by a human or LLM scientist. This brings transparency to the critique process.

While these summary statistics can highlight potential discrepancies, we need a method to determine whether these discrepancies are meaningful. To address this, we show how we can automatically convert the summary statistics produced by CALM into hypothesis tests, for many commonly-used scientific models. Specifically, if we can sample from the scientific model (Gelman et al., 2013; Cranmer et al., 2019), we can form a null distribution for a summary statistic and compute an empirical p-value. Thus, we can transform each summary statistic into a quantitative check, providing a rigorous way to assess both the significance of the discrepancies and the validity of the model. In doing so, we reduce the complex task of automatically validating proposed models and critiques to the well-understood problem of hypothesis testing. We can view these quantitative checks as (loosely) serving a role analogous to how formal verification systems validate proofs in LLM-based theorem proving systems like AlphaProof (DeepMind, 2024).

In experiments (Section 4), we evaluate CALM along key qualitative and quantitative properties crucial for an automated critic system. In settings where we synthetically control discrepancies between models and datasets, CALM consistently identifies true discrepancies and avoids hallucinating false ones. We also assess important qualitative aspects of CALM's critiques (*e.g.,* transparency), and find that both LLM and human judges prefer CALM's critiques over alternatives. Finally, we demonstrate the practical impact of CALM's critiques on the downstream task of guiding an LLM-based scientific model discovery system. On real-world datasets, CALM's critiques enable an LLM-based automated model discovery system (Li et al., 2024) to significantly improve upon initial human-designed models.

## 2 BACKGROUND

In this section, we describe model criticism techniques, from different domains, that are commonly used to find discrepancies. Crucially, we can often formalize finding discrepancies as identifying suitable *test statistics*, using those statistics to compute discrepancies between model predictions and data, and validating their *significance* using domain knowledge.

**Regression analysis** In regression analysis, we begin with a dataset $\mathcal{D} = \{\mathcal{X}, \mathcal{Y}\}$ of input features $\mathcal{X}$ and targets $\mathcal{Y}$; our goal is to predict $\mathcal{Y}$ from $\mathcal{X}$. Given model predictions $Y^{\text{pred}}$, we perform *model diagnostics* that target the standard assumptions of linear regression (*e.g.,* linearity, homoscedasticity, uncorrelated errors). For example, to evaluate whether homoscedasticity holds, we can plot the residuals against the input features. We can then either informally assess whether the pattern in the residuals indicates a significant departure from homoscedasticity or perform statistical tests.

**Computational models** Computational models often make simplifying assumptions that can lead to systematic errors, even after the parameters of these models are calibrated. This might be due to imperfect physical knowledge or systematic measurement errors; these systematic errors are often known as *model inadequacies*. Bayarri & Berger (2000) introduce a framework for understanding these inadequacies that involves defining domain-specific evaluation criteria or performing sensitivity analyses and checking whether these accord with scientific intuition. Another very influential approach is to cast this as a statistical modeling problem and directly build a statistical model of the discrepancy (Kennedy & O'Hagan, 2001). Building on this work, Joseph & Yan (2015) show how to study this discrepancy through an analysis of variance decomposition.

**Bayesian statistical models** In statistical modeling, we model the data as a probability distribution. More formally, a statistical model defines a joint probability distribution $p(\mathcal{Y}, \theta | \mathcal{X}, \mathcal{H})$ over observed variables $\mathcal{Y}$ and latent variables $\theta$; we use $\mathcal{H}$ to indicate a specific class of statistical models and the dataset $\mathcal{D} = \{\mathcal{X}, \mathcal{Y}\}$ can include both observations $\mathcal{Y}$ that we model as random variables and additional quantities $\mathcal{X}$ that we treat as fixed. By marginalizing out the latent variables, we

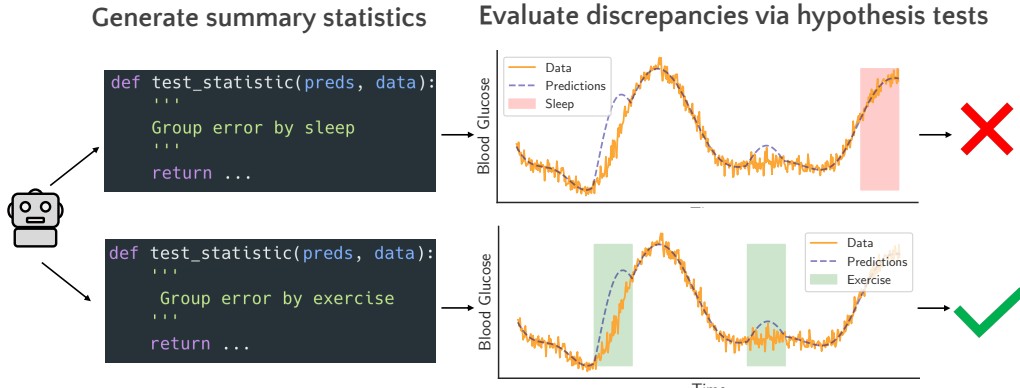

Figure 1: **Criticizing scientific models with CALM.** First, an LLM generates *summary statistics* that capture potential discrepancies that are tailored to the model and dataset; the LLM conditions on dataset metadata and a symbolic representation of a scientific model. We use these summary statistics to perform *hypothesis tests* to evaluate the *significance* of each discrepancy.

obtain the *posterior predictive distribution*

$$p(Y^{\text{ppred}} \mid \mathcal{D}, \mathcal{H}) = \int p(Y^{\text{ppred}}|\theta, \mathcal{H})p(\theta|\mathcal{D}, \mathcal{H})d\theta \tag{1}$$

A common technique for evaluating such a model is a *posterior predictive check* (PPC) (Box, 1980; Gelman et al., 1996; Meng, 1994; Rubin, 1984). In brief, PPCs ask if the posterior predictive distribution captures important properties of the data. Concretely, to perform a PPC, we first draw samples from the posterior predictive distribution, $\{Y_i^{\text{ppred}}\}_{i=1}^m \sim p(Y^{\text{ppred}} \mid \mathcal{D}, \mathcal{H})$. We then choose a *test statistic* $T(\mathcal{X}, Y^{\text{ppred}})$ that can reveal some *property* of the data that is not well-captured by the model samples. To compare the posterior predictive samples against the dataset, we compute the test statistic over both samples (forming a null distribution) and data. For a PPC to be useful, the test statistic must be chosen in a model-dependent way and choosing an appropriate test statistic is an important step in many applied modeling settings (van Dyk & Kang, 2004; Belin & Rubin, 1995; Gelman et al., 2005). For example, when criticizing a Poisson model, one might check for over-dispersion by computing the variance-to-mean ratio. Crucially, posterior predictive checks do not require human intervention, since they automatically generate a quantitative measure of the significance of any discrepancy via the posterior predictive p-value; we discuss this in more detail in Section 3.1.

## 3 METHOD: CALM

In this section, we describe CALM, our system for finding systematic discrepancies between a scientific model and dataset. We provide a brief overview here; for a schematic overview, see Figure 1. CALM takes as input: dataset metadata, a symbolic representation of a model (*e.g.,* program) and model samples. Given these, CALM produces significant discrepancies. Each discrepancy is represented as a *test statistic* implemented as a `Python` function, an executable artifact that programmatically expresses the discrepancy, and a *natural language criticism*.

### 3.1 AUTOMATICALLY PROPOSING AND EVALUATING DISCREPANCIES

**Proposing discrepancies via test statistics** As we saw in Section 2, we can often formalize finding discrepancies between model predictions and data as identifying suitable test statistics. Designing test statistics that capture systematic discrepancies between a model and dataset requires modeling expertise, domain knowlege, and strong programming capabilities. We use LLMs to automate this process. To propose test statistics, the LLM conditions on dataset metadata, $\mathcal{C}$ (*e.g.,* description of the dataset, column names) and a symbolic representation of a model $\mathcal{H}$; for examples of these inputs, see Figure 9. To implement these test statistics, LLMs write Python functions that

---

**Algorithm 1:** Producing test statistics and empirical p-values

---

**Input:** dataset $\mathcal{D}$, metadata $\mathcal{C}$, model $\mathcal{H}$, model samples $\{Y_i^{\mathrm{pred}}\}_{i=1}^m$, number proposals $n$

$\{T_k\}_{k=1}^n \sim p_{\mathrm{LM}}(\cdot|\mathcal{C}, \mathcal{H})$

$\{p_k\}_{k=1}^n \leftarrow \texttt{get-empirical-pval}(\mathcal{D}, \{Y_i^{\mathrm{pred}}\}_{i=1}^m, \{T_k\}_{k=1}^n)$ via Equation 2

$\{\tilde{p}_k\}_{k=1}^n = \texttt{multiple-test-adjustment}(\{p_k\}_{k=1}^m)$

**Output:** test statistics $\{T_k\}_{k=1}^m$, adjusted empirical p-values $\{\tilde{p}_k\}_{k=1}^m$

---

take a `pandas` dataframe as input; the dataframe contains $\mathcal{X}$ and either the data $\mathcal{Y}$ or a model sample $Y_i^{\mathrm{pred}}$ of the same dimension. By design, these `Python` functions can be easily executed and inspected by a human or LLM scientist and, as we will experimentally validate, help improve transparency. For examples of the functions produced by CALM, see Sections A.5 and A.6. For our test statistic proposer, we use `gpt-4-turbo-2024-04-09`. See Figure 7 for the prompt.

**Evaluating significance of discrepancies via hypothesis tests**   We now describe how CALM uses the test statistics to identify significant discrepancies. In brief, we use model samples to approximate a null distribution over the test statistic and then compute an empirical p-value. We assume the user can simulate data from the model $\{Y_i^{\mathrm{pred}}\}_{i=1}^m$. This is not restrictive requirement and how the user generates the model samples is a design choice; for example, we can do this for any model that describes a generative process for the data.

We describe how to estimate an empirical p-value $p_k$ given $T_k$ and $\{Y_i^{\mathrm{pred}}\}_{i=1}^m$ below.

1. We approximate the null distribution of the test statistic by computing the test statistic over the model samples $\{T(\mathcal{X}, Y_i^{\mathrm{pred}})\}_{i=1}^m$.

2. We locate the test statistic of the observed data $T(\mathcal{X}, \mathcal{Y})$ within this null distribution to obtain an empirical p-value. That is, we compute

$$P(T(\mathcal{X}, Y^{\mathrm{pred}}) \geq T(\mathcal{X}, \mathcal{Y})|\mathcal{D}, \mathcal{H}) \approx \frac{1}{m} \sum_{i=1}^m \mathbb{1}_{\{T(\mathcal{X}, Y_i^{\mathrm{pred}}) \geq T(\mathcal{X}, Y)\}} \tag{2}$$

We visualize the computation of the p-values in the Appendix (Figure 10). To capture different discrepancies, we compute multiple test statistics in parallel for a model-dataset pair. However, this can inflate the effective false positive rate: for large enough $m$, we expect $\min_k p_k \leq \alpha$ even if the model and dataset have no discrepancy. We thus apply a Bonferroni correction to obtain adjusted p-values $\{\tilde{p}_k\}_{k=1}^m$. We regard all $T_k$ such that $\tilde{p}_k \leq \alpha$ as significant.

**Instantiating the framework for Bayesian models**   In our experiments, we focus our evaluation on Bayesian models because they are widely used in scientific settings (Gelman et al., 2013; Cranmer et al., 2019). In our context, Bayesian models are also appealing because they can be expressed symbolically as probabilistic programs (van de Meent et al., 2021; Goodman, 2013) and we can choose the model samples to be posterior predictive samples $\{Y_i^{\mathrm{ppred}}\}_{i=1}^m$ (Equation 1). The corresponding *posterior predictive p-value* has an intuitive interpretation: how atypical is $\mathcal{Y}$ under the posterior distribution $p(Y^{\mathrm{ppred}}|\mathcal{D}, \mathcal{H})$ with respect to the discrepancy measure defined by $T_k$?

## 3.2 INTERFACING WITH LLM SCIENCE AGENTS VIA NATURAL LANGUAGE CRITICISM

In many situations, we might want to integrate CALM within a broader scientific discovery system, involving either human or LLM scientists. Therefore, CALM also produces *natural language criticism*. This design choice is motivated by several considerations. By offering critiques in natural language, which is flexible and generic, the system provides an additional medium for users to interpret results, which can be useful in fields where training in formal modeling is less common. Second, this design choice is natural given recent advances in LLM based agents for scientific discovery and modeling (Huang et al., 2024; Li et al., 2024).

We prompt an LLM to produce *natural language criticism* $h_k$ that summarizes the discrepancy implied by test statistic $T_k$ and its p-value $\tilde{p}_k$. Specifically, we ask the LLM to synthesize the test

statistic in a way that's informative to a colleague revising some initial model. For examples of the natural language critiques produced, see Section A.4 and for the prompt see Figure 8.

We can easily integrate these three artifacts within an LLM based scientific discovery system. Specifically, we provide the system with (1) a `Python` implementation of the test statistic $T_k$, (2) the natural language $h_k$, and (3) the initial model; in our experiments these models will be probabilistic programs in `pymc` or `stan` (Carpenter et al., 2017; Abril-Pla et al., 2023). We use the LLM-based system for generating probabilistic programs introduced by Li et al. (2024).

In general, these hypothesis tests are cheap relative to the cost of model fitting. For example, posterior inference is the dominating cost for Bayesian models and performing posterior predictive checks are cheap given posterior samples. Thus, CALM will generally introduce minimal overhead to the overall cost of an AI scientist system.

## 4 EXPERIMENTS

In this section, we present experimental results that evaluate key quantitative and qualitative properties of our system. We begin by illustrating the pitfalls of a naive LLM in a synthetic regression setting. We then systematically study CALM's ability to avoid hallucinations and discover true discrepancies by analyzing its true and false positive rates in a setting where we synthesize discrepancies between models and datasets. We then evaluate the *transparency* and *interpretability* of our system in human user and LLM evaluations and the *actionability* of the natural language criticism in helping an LLM-based system to revise models.

### 4.1 EXPERIMENT 1: NAIVE LLM-BASED CRITIC HALLUCINATES IN SYNTHETIC REGRESSION TASK

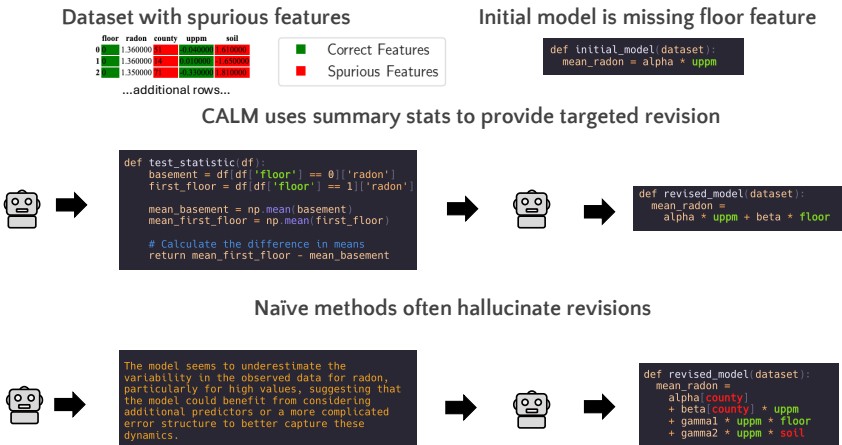

Figure 2: **Illustrating how CALM avoids hallucinated revisions.** CALM hypothesizes discrepancies via *summary statistics* and makes *targeted* changes to the initial model, which is missing the feature `floor`. In contrast, the naive method hallucinates (see explanation for details) and introduce spurious features (*e.g.,* `county`, `soil`) to the initial model. We highlight spurious features in red and correct features in green in code.

In an initial case study, we show that a naive LLM critic consistently hallucinates but CALM does not, in a synthetic regression setting. Specifically, we characterize the model revision changes induced by the critiques produced by CALM and the naive approach, in a setting where we adversarially introduce spurious "distractor" features into a dataset. For an overview, see Figure 2.

**Generating a regression dataset with spurious features**     We generate a synthetic dataset inspired by the `radon` dataset, a commonly used dataset in regression analysis. We generate the target, `radon` as a a linear function of `floor` (basement or first floor) and `uppm` (*e.g.,* uranium), corrupted

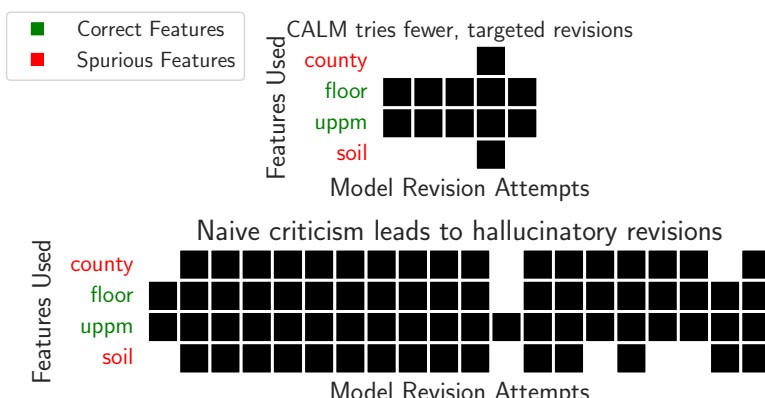

Figure 3: **CALM attempts more fewer, more targeted revisions.** The critiques produced by the naive approach drive greedy model revisions that indiscriminately add both spurious (red) and correct (green) features; we indicate features used in revised models as dark-colored squares. In contrast, CALM leads to fewer revisions because it filters discrepancies by significance. Furthermore, those revisions generally target the correct missing feature (`floor`).

with additive Gaussian noise. In addition to these two features, we add two additional *spurious*, distractor features to the dataframe, `county` and `soil`, with semantically plausible names.

**Naive LLM critic baseline**   We implement a naive approach to model criticism that receives (1) an initial statistical model represented as a `pymc` program (2) a dataframe of the posterior predictive mean radon predictions along with the corresponding variances of those predictions and a (3) dataframe of the dataset. Given this information, we ask the LLM critic to identify discrepancies between the predictions and data.

**Evaluating critiques in driving model revision**   We generate twenty critiques from both CALM and the naive baseline; the initial model regresses radon against only `uppm`, omitting `floor` which is used in the ground truth. CALM filters the critiques to five significant ones ($p < 0.01$); four correctly identify that `radon` varies by floor, and the other correctly notes that model fails to capture the range of `radon` values but does not identify that the missing `floor` feature is the culprit. In contrast, the naive approach recommends generic model expansions; for an example see the text in the bottom of Figure 2. We evaluate the critiques by feeding them into an LLM-based model revision system (Li et al., 2024) as described in Section 3.2. Figure 3 shows the features added per revision attempt, with spurious features indicated in red and correct features in green. Rows correspond to features and columns correspond to model revision attempts; we indicate that a feature was added using dark-colored squares. The naive approach often adds all possible features indiscriminately. In contrast, the majority of CALM's critiques lead to targeted revisions. The main exception is the critique about the discrepancy in the range of values; this critique captures a true deficiency in the initial model, but isn't actionable (*i.e.,* suggest a concrete strategy for revising) which leads the revision LLM to be greedy; we will evaluate this notion of actionability in additional experiments.

### 4.2   EXPERIMENT 2: STATISTICAL ANALYSIS OF HALLUCINATIONS AND TRUE DISCOVERIES

A reliable critic system should *avoid hallucinations* (*i.e.,* generating false positives) and *discover true discrepancies* when they exist. In this section, we study this through a statistical lens and characterize CALM's false and true positive rates. We ask: does CALM reliably *discover* discrepancies when there are actual discrepancies? And, conversely, does CALM hallucinate when there are no discrepancies? To study this, we synthetically generate discrepancies between models and datasets which enables us to empirically characterize the true and false positive rates.

**Generating no-discovery and discovery datasets**   We synthetically generate model-dataset pairs where each pair is either a *no-discovery* pair or *discovery* pair.

To construct a *no-discovery* pair, we first sample a dataset $\mathcal{Y}$ from a ground truth data distribution $\mathcal{Y} \sim p(\mathcal{Y}|\mathcal{H})$. We then draw $m$ posterior predictive samples from the ground truth data distribution

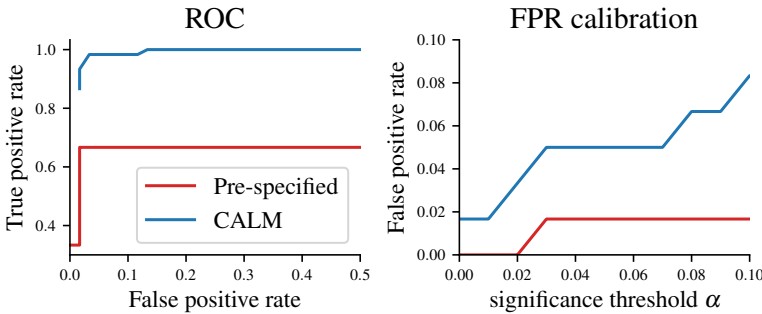

Figure 4: **Statistical analysis of CALM's ability to discover discrepancies and avoid hallucinations.** (**left**) True positive rate (TPR) vs. false positive rate (FPR) at different significance thresholds. (**right**) FPR against significance threshold. CALM correctly identifies more discrepancies than the pre-specified method, at the same FPR level. The FPR is calibrated with the significance threshold, showing that CALM systematically avoids hallucinations.

conditioned on the data: *i.e.,* $\{Y_i^{\text{ppred}}\}_{i=1}^m \sim p(Y^{\text{ppred}}|\mathcal{Y}, \mathcal{H})$. No-discovery pairs serve as a negative control to ensure that CALM does not systematically hallucinate and produce false discoveries.

To generate *discovery* pairs, we sample a dataset $\mathcal{Y}$ from a dataset distribution $p$. However, we pair $\mathcal{Y}$ with samples from a *lesioned* model $q$, where we choose $q$ so that it fails to capture an important aspect of the data generating distribution $q$. For example, we can take $p$ to be a Student's t distribution and $q$ to be a Gaussian distribution; even after conditioning on data $q(Y|\mathcal{Y})$ will fail to capture the tails. These discovery pairs serve as a positive control and allow us to understand how reliably CALM identifies discoveries (*i.e.,* true positive rate).

We generate six model-dataset pairs. The data-generating models are: Student's t, negative binomial, and a generalized linear model. The lesioned models are: Gaussian, Poisson, and logistic growth. To account for randomness in data generation, we generate twenty copies of each model-data pair corresponding to twenty random fresh datasets.

**Calculating true positive and false positive rate** For each model-dataset pair, we run CALM with 24 proposals and at a temperature 0.7. Our system decides if there is a discrepancy by checking whether the minimum p-value is less than the significance threshold; that is, whether $\min_k \tilde{p}_k \leq \alpha$. By construction, we have the "correct" decision for each pair. To compute the true positive rate, we compute the proportion of discovery pairs in which CALM correctly decided there was a discrepancy. To compute the false positive rate, we compute the proportion of no-discovery pairs in which CALM incorrectly decided there was a discrepancy.

**Quantitative Results** In Figure 4, we show the true and false positive rates of CALM. As a baseline, we compare CALM against a standard set of pre-specified test statistics: mean and variance. From the ROC curve, we see that CALM exhibits a favorable trade-off between the true positive rate (power) and false positive rate (type I error), and significantly outperforms the baseline method, achieving a higher true positive rate at all false positive rate levels. As the false positive rate (FPR) calibration plot shows, the false positive rate closely tracks the significance threshold $\alpha$, showing that our system does not systematically identify spurious discrepancies. These analyses illustrate that CALM has favorable statistical properties in a controlled setting. In the appendix (Section A.5), we show examples of CALM's proposed test statistics that account for its favorable statistical properties. These statistics are tailored to the statistical model. For example, CALM proposes kurtosis for the Student's t setting, to assess the tails of the distribution.

### 4.3 EXPERIMENT 3: ANALYZING KEY QUALITATIVE PROPERTIES OF TEST STATISTICS FOR REAL-WORLD MODEL-DATASET PAIRS

In the previous sections, we evaluated CALM's statistical properties. However, users interacting with an LLM-based critic system may care just as much about key *qualitative* properties such

as *transparency* (*i.e.,* how clear is the reasoning used to generate the critique) and *actionability* (*i.e.,* how useful is the criticism to a scientist revising the model).

To study this, we first apply CALM to real world datasets and expert written models covering a range of scientific domains (see Section A.3). We quantitatively assess CALM-generated critiques in both human and automated LLM-based evaluation. We then qualitatively characterize CALM's critiques, which illustrate the conceptual advantages of using CALM's tailored test statistics.

**Experimental Setup**   The `Stan PosteriorDB` database (Magnusson et al., 2023) consists of real-world datasets and probabilistic models implemented in `Stan`; these models are open-source contributions from the `Stan` developer community that cover a broad range of modeling motifs ranging from hierarchical modeling to regression. We chose 36 model-dataset pairs based on ones used in several recent papers (Modi et al., 2023; Li et al., 2024; Wu & Goodman, 2022). For each StanDB model-dataset pair, the LLM proposes twenty-four test statistics $\{T_k\}_{k=1}^{24}$; we run this proposal step at a temperature 1.0. Then, for each $T_k$, we generate natural language criticism $h_k$ by running the natural language criticism step at a temperature 0.0.

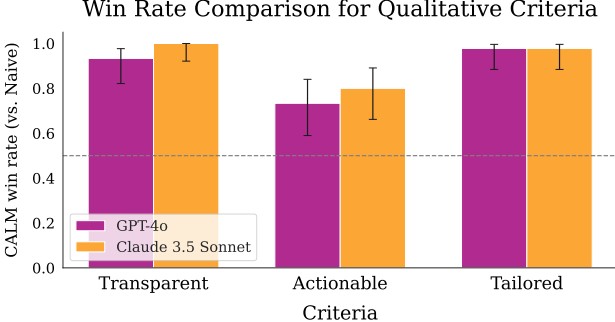

Figure 5: **CALM criticisms have higher win rates versus naively generated criticisms.** Critiques are rated on three qualitative criteria by LLM-based judges (GPT-4o and Claude 3.5 Sonnet). LLM-based judges are aligned with human evaluators: GPT-4o and Claude 3.5 Sonnet have 100% alignment for transparent and tailored preferences, and are 80% and 90% aligned for actionable preferences, respectively. Error bars represent 95% confidence intervals (Wilson score).

**Systematic evaluation of qualitative properties of critiques**   We conducted a human evaluation study with three Ph.D. students (non-authors) with expertise in statistics, who were blind to the critic methods. We randomly selected ten model-dataset pairs. For each pair, both CALM and a naive LLM critic generated critiques. The evaluators chose which critique was better along three criteria that we describe and motivate below.

1. **Transparency**: can a user of the system understand how the critique was produced? Transparency is important for building trust with users and can also help them evaluate if the system is hallucinating.

2. **Actionable**: can the critique help a scientist revise the model? A critique is more useful if it provides insights into how to revise the model. For example, knowing that a model has high error is less useful than knowing that a model has high error on a specific sub-population.

3. **Tailored**: is the critique targeted for the specific model and dataset? We do not expect generic critiques to provide much insight.

To scale this analysis, we employed state-of-the-art LLM-based judges (`gpt-4o-2024-08-06` and `claude-3-5-sonnet-20240620`) following highly specific guidelines; for details, see the Appendix A.8. In Figure 5, we show the win-rates (higher is better) across two LLM judges and the three criteria. CALM is classified as significantly more transparent ($\sim 97\%$), actionable ($\sim 76\%$), and tailored ($\sim 98\%$). Both LLM-based judges are aligned with human preferences, having 100% alignment for transparent and tailored preferences, and GPT-4o and Claude 3.5 Sonnet having 80% and 90% alignment for actionable preferences, respectively. The domain experts gave qualitative feedback in support of CALM's approach, particularly in terms of transparency. Experts noted the benefit of immediately executable code for quick assessment (see Appendix A.7 for quotes).

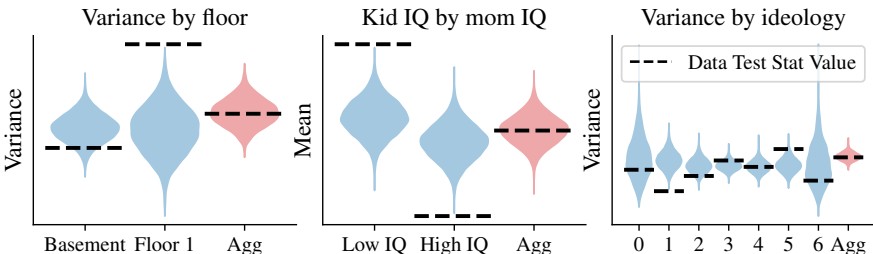

Figure 6: **CALM proposes test statistics that slice model predictions based on input.** Each violin is a test statistic distribution of model samples on a slice (*e.g.,* variance of model predictions for first floor measurements). The horizontal lines indicate test statistics computed on data. We indicate the test statistic type on the y-axis and slice category on the x-axis; the "agg" slices indicates aggregation across all slices. Blue violins correspond to sliced test statistics and red violins correspond to aggregated ones. The dashed line passes through red violin centers but not the blue ones, showing that CALM's choice to slice test statistics reveals discrepancies that pre-specified ones cannot.

**Qualitative examples: sliced test statistics**   One specific kind of test statistic that meets the above three criteria are *sliced test statistics*. CALM often proposes test statistics that slice the model prediction $Y_i^{\text{pred}}$ based on the input features $\mathcal{X}$. For example

```python
1  # Filter to get basement and non-basement samples
2  basement_samples = df[df['floor_measure'] == 0]['y_rep']
3  non_basement_samples = df[df['floor_measure'] == 1]['y_rep']
4
5  # Compute standard deviations for both subsets
6  std_basement = np.std(basement_samples)
7  std_non_basement = np.std(non_basement_samples)
8
9  # Compute the difference in standard deviations as the test statistic
10 test_statistic_value = std_basement - std_non_basement
```

In Figure 6, we illustrate the benefits of CALM's sliced test statistics over pre-specified, aggregate test statistics. We compare test statistic distributions computed from model samples against the data, sliced by the input values (*e.g.,* variance of model predictions for radon measurements in basement vs first floor). Sliced test statistics reveal discrepancies that the aggregated ones cannot. We provide additional randomly-sampled test statistics in the Appendix (Section A.6).

### 4.4   EXPERIMENT 4: CALM GENERATED CRITICISM DRIVES MODEL IMPROVEMENTS

| Method | Wins > 1 SE | Wins > 1.5 SE | Wins > 2.0 SE |
|---|---|---|---|
| CALM | 0.94 | 0.94 | 0.82 |
| Initial Model | 0.06 | 0.06 | 0.06 |

Table 1: **CALM consistently improves over the initial model.** CALM achieves significantly higher win rates compared to the initial model at various standard error (SE) thresholds. We say a win is significant at a given SE threshold if the difference in scores is larger than the SE margin.

| Method | Wins > 1 SE | Wins > 1.5 SE | Wins > 2.0 SE |
|---|---|---|---|
| CALM | 0.59 | 0.59 | 0.53 |
| Data-blind | 0.29 | 0.29 | 0.29 |

Table 2: **CALM outperforms data-blind method.** CALM demonstrates higher win rates across all standard error (SE) margins compared to data-blind method that conditions only on the symbolic representation of the model.

Model criticism should ideally be *actionable* and aid a user (either LLM or human) in model revision. In our final experiment, we use the model criticism generated by CALM in the previous section

(4.3) to aid an LLM-based agent in revising an initial model. We show that CALM's critiques lead to significant improvements over the initial model.

**Experimental Setup** We integrate CALM into the model discovery system introduced by Li et al. (2024) by giving a *revision-LLM* three components: the initial model $\mathcal{H}$ implemented as a probabilistic program, the test statistic $T_k$, and natural language criticism $h_k$; the criticism was produced in the previous section by running CALM on the (fitted) initial model. We fit the models proposed by the revision LLM using `pymc` (Abril-Pla et al., 2023); we repeat each proposal three times at a temperature 0.0 since we noticed non-determinism. We allow the LLM to revise based on a filtered set of significant test statistics. For details, see Section A.9. We report the best model across proposals and test statistics.

**Ablation** We consider a *data-blind* LLM critic that receives only the statistical model, implemented as a probabilistic program; we run this at a temperature 0.0. This approach can be effective since modeling assumptions are enumerated in the initial probabilistic program.

**Quantitative Results** In Tables 1 and 2, we evaluate CALM's ability to produce critiques that improve upon an initial statistical model and outperform the data-blind method. To do this, we first compute the expected log predictive density (ELPD LOO) score for both initial and revised models (Vehtari et al., 2017). Next, we calculate the score difference between the initial and revised model and determine the *margin* of victory by dividing the score difference by the standard errors (SE) of the score difference. For a given dataset, a method is considered to "win" over another at a specific SE margin if the score difference is larger than the margin. For example, if the score difference is 2 and the SE margin is 1, we count it as a significant win. Finally, we compute aggregated win rates at various SE margins (1, 1.5, 2). The win rate is the percentage of datasets where one method outperformed another at a given SE margin.

CALM's critiques help a revision LLM significantly improve upon the initial model over $80\%$ of the time, which shows that CALM reliably produces *actionable* critiques. CALM's successes are often related to sliced statistics discussed in Section 4.3 (*e.g.,* the revision-LLM introduces floor-dependent variance terms). Furthermore, in Table 2, we show that CALM also outperforms the data-blind critic. Next, we discuss CALM's limitations.

**Limitation 1: Suboptimal transformations of data** CALM does not see the model predictions or data. As a consequence, CALM sometimes does not provide good critiques on *transforming data*. This happens most prominently in the `mesquite` setting where the model predictions can be negative even though the data is non-negative.

**Limitation 2: Correct criticism but imperfect implementation** In some cases, CALM identifies a legitimate discrepancy that the revision LLM incorrectly implements; the revision LLM correctly uses a Categorical likelihood but does not transpose the logits correctly.

## 5 CONCLUSION

We introduced CALM, a framework for automated model criticism that leverages LLMs to identify discrepancies between a model and dataset and then applies hypothesis tests to assess the significance of discrepancies. CALM serves as a lightweight verifier, validating both scientific models and critiques within a hypothesis testing framework. Our experiments demonstrate that CALM reliably identifies true discrepancies without hallucinating false critiques. Futhermore, both human and LLM judges preferred CALM's critiques over alternative approaches. CALM critiques enabled an LLM-based system to substantially improve upon expert designed models. By automating model criticism, CALM represents a step toward more reliable automatic scientific discovery systems.

While our evaluation was limited to Bayesian models, which are commonly used in scientific domains, CALM's design is versatile: the only requirements are the ability to simulate data from the model and a symbolic representation of the model. An exploration of other common classes of scientific models (Cranmer et al., 2019) is an exciting direction for future work.

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

# A APPENDIX

## A.1 PROMPTS/INPUTS FOR TEST STATISTIC PROPOSER STEP

---

**Critic function prompt**

```
You are a brilliant statistician specializing in critiquing models!  Your equally
brilliant colleague has come up with a probabilistic program in Stan that proposes a
generative/statistical model for the data.
Your job is to critique the models and provide hypotheses for discrepancies between
the model and the data.  To do this, you should write a "test statistic function"
in Python.  This is motivated by posterior predictive checks in Bayesian statistics.
This test function should take as input a dataframe where one of the columns contains
the posterior predictive sample.  It should return a scalar-valued test statistic.
To quantify discrepancies, I will compute this test statistic for each posterior
predictive sample and compare it to observed data.  Therefore, choose test statistics
that you think will reveal discrepancies between the model and the data.
```

---

Figure 7: **System prompt for proposing test statistics** Prompt used for proposing test statistic step in Section 3.1. We also provide some additional instructions on formatting the response and describing the format of the input dataframe which we omit from the prompt above.

---

**Natural language criticism prompt**

```
Your equally brilliant colleague has come up with a discrepancy functions that identify
possible weaknesses of generative models for data.  I will give you the test statistics
and the result of computing those test statistics.  Your job is to interpret the
results of running those test statistics and synthesize the discrepancies.  Your
synthesis should be as helpful as possible for your colleague who will use this
synthesis to improve the model.  You will be given one million dollars if you do
this well.  Focus on being as informative with your synthesis (do not say generic
things) to help your colleague understand the test statistic.  You should provide a
natural language summary of the discrepancy function.  Reference specifically the test
statistic type and the discrepancy it reveals about specific modeling assumptions.  I
provide the test statistic Python function and posterior-predictive pval.

Posterior predictive p-val:
Test statistic function:
```

---

Figure 8: **Prompt for natural language criticism step** See Section 3.2.

## A.2 COMPUTING P-VALUES

## A.3 STAN POSTERIORDB DATASETS

We list the model-dataset pairs criticized in Section 4.3.

- radon_mn-radon_variable_slope_noncentered
- radon_mn-radon_variable_intercept_slope_centered
- radon_mn-radon_partially_pooled_noncentered
- radon_mn-radon_county_intercept
- radon_mn-radon_pooled
- radon_mn-radon_variable_intercept_centered
- radon_mn-radon_variable_intercept_slope_noncentered
- radon_mn-radon_variable_intercept_noncentered
- radon_mn-radon_partially_pooled_centered
- radon_mn-radon_variable_slope_centered
- kidiq-kidscore_momhs
- kidiq-kidscore_momiq
- kidiq_with_mom_work-kidscore_interaction_z
- kidiq_with_mom_work-kidscore_interaction_c

**Inputs to test statistic proposer**

```
1  Description: Cognitive test scores of three and four-year-old
       children
2
3  Column Description:
4
5    - kid_score: cognitive test scores of three and four-year-old
       children
6
7    - mom_hs   : did mother complete high school? 1: Yes, 0: No
8
9    - mom_iq   : mother IQ score
10
11 data {
12   int<lower=0> N;
13   vector<lower=0, upper=200>[N] kid_score;
14   vector<lower=0, upper=1>[N] mom_hs;
15 }
16 parameters {
17   vector[2] beta;
18   real<lower=0> sigma;
19 }
20 model {
21   sigma ~ cauchy(0, 2.5);
22   kid_score ~ normal(beta[1] + beta[2] * mom_hs, sigma);
23 }
```

Figure 9: **Examples inputs to test statistic proposal step**. Contextual information provided to test statistic proposer in Section 3.1. Programs are implemented in Stan. Dataset metadata was available with the dataset.

- kidiq_with_mom_work-kidscore_mom_work
- kidiq_with_mom_work-kidscore_interaction_c2
- GLM_Poisson_Data-GLM_Poisson_model
- dugongs_data-dugongs_model
- eight_schools-eight_schools_centered
- surgical_data-surgical_model
- nes1972-nes
- gp_pois_regr-gp_pois_regr
- earnings-logearn_height_male
- earnings-log10earn_height
- earnings-earn_height
- earnings-logearn_interaction
- earnings-logearn_interaction_z
- earnings-logearn_height
- mesquite-mesquite
- mesquite-logmesquite_logvolume
- mesquite-logmesquite_logvash
- mesquite-logmesquite_logvas
- mesquite-logmesquite

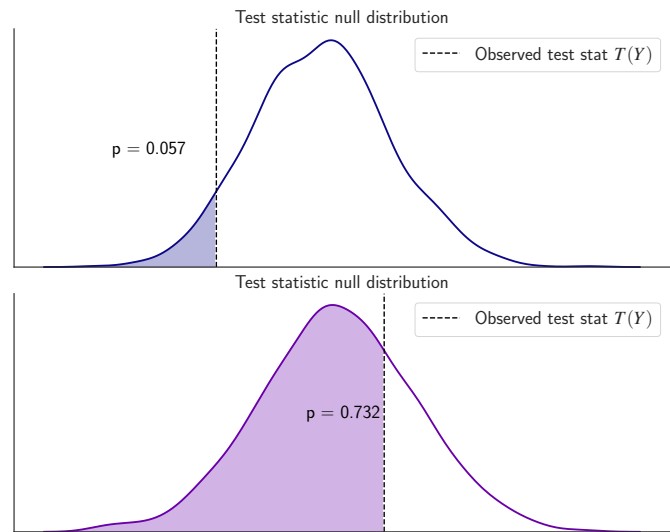

Figure 10: **Computing empirical p-values from test statistics.** We illustrate how to use test statistics to obtain p-values. In short, we compute the test statistic over the model samples to form a null distribution over the test statistic; we do this for two different test statistics above. We then compute the test statistic on the data (observed test stat in the plots) which appear as vertical dashed lines in the figure. We then compute an empirical value by comparing the observed test statistic value against the null distribution.

- `low_dim_gauss_mix-low_dim_gauss_mix`
- `low_dim_gauss_mix_collapse-low_dim_gauss_mix_collapse`
- `arK-arK`

### A.4    EXAMPLES OF NATURAL LANGUAGE CRITIQUES

```
1  """
2  The posterior predictive p-value of 0.0 suggests a significant
       discrepancy between the observed data and the model predictions
       regarding the distribution of children's cognitive test scores. The
       test statistic used here measures the skewness of the predicted
       scores, indicating that the model's assumption of normally
       distributed scores may not be appropriate.
3
4  The very low posterior predictive p-value of 0.0019 suggests that the
       model does not adequately capture the variability of radon levels
       across different floor measurements. This discrepancy indicates that
       the assumption of a homogeneous linear interaction between floor
       level and radon level across all counties may be too simplistic.
5  """
6
7  """
8  The model fails to capture the difference in radon levels between
       measurements taken in the basement versus the first floor. This is
       indicated by a posterior predictive p-value of 0.0, suggesting that
       the model does not adequately represent the known higher radon levels
        typically found in basements compared to the first floor.
9  """
10
11 """
12 The model's use of a normal distribution to predict the inherently
       discrete variable 'partyid7' (ranging from 1 to 7) results in a
       significant proportion of predicted values falling outside this
       permissible range.
```

```
13 """
```

## A.5 EXAMPLE TEST STATISTICS FOR EXPERIMENT 2

```
1 def test_statistic():
2     kurtosis = (fourth_moment / squared_var) - 3  # excess kurtosis
3
4 def test_statistic():
5     # Calculating the first derivative (approximated by finite
      differences)
6     derivatives = population_diff / year_diff
7     positive_slope_count = np.sum(derivatives > 0)
8
9 def test_statistic():
10     dispersion_ratio = y_rep_variance/y_rep_mean if y_rep_mean != 0 else
      float('inf')
```

## A.6 FURTHER TEST STATISTICS FOR EXPERIMENT 3

```
1 def test_statistic(df):
2     quantile_edges = df['log_uppm'].quantile([0.33, 0.66]).tolist()
3     def categorize_by_uranium(uppm):
4         if uppm <= quantile_edges[0]:
5             return 'Low uranium'
6         elif uppm <= quantile_edges[1]:
7             return 'Medium uranium'
8         else:
9             return 'High uranium'
10     df['uranium_category'] = df['log_uppm'].apply(categorize_by_uranium)
11     df['residuals'] = df['y_rep'] - df['y_rep'].mean()
12     grouped_std_devs = df.groupby('uranium_category')['residuals'].std()
13     range_of_std_devs = grouped_std_devs.max() - grouped_std_devs.min()
14
15 def test_statistic(df):
16     grouped_variances = df.groupby('county_idx')['y_rep'].var()
17     test_statistic_value = np.std(grouped_variances)
18
19 def test_statistic(df):
20     group_0 = df[df['group'] == 0]['y_rep']
21     group_1 = df[df['group'] == 1]['y_rep']
22
23     std_dev_group_0 = np.std(group_0)
24     std_dev_group_1 = np.std(group_1)
25
26     diff_std_dev = abs(std_dev_group_0 - std_dev_group_1)
27
28 def test_statistic(df):
29     range_per_county = df.groupby('county_idx')['y_rep'].apply(lambda x:
      x.max() - x.min())
30     average_range = range_per_county.mean()
31
32 def test_statistic(df):
33     iq_bins = pd.cut(df['mom_iq'], bins=[0, 90, 100, 110, 120, 130, np.
      inf], right=False, labels=False)
34     variances_by_iq_range = df.groupby(iq_bins)['y_rep'].var()
35     coefficient_of_variation = variances_by_iq_range.std() /
      variances_by_iq_range.mean()
36
37 def test_statistic(df):
38     test_statistic_value = np.var(df['y_rep'])
39     return result
40
41 def test_statistic(df):
```

```
42      males_log_earn_rep = df[df['male'] == 1]['y_rep']
43      females_log_earn_rep = df[df['male'] == 0]['y_rep']
44
45      std_dev_male = np.std(males_log_earn_rep)
46      std_dev_female = np.std(females_log_earn_rep)
47
48      test_statistic_value = abs(std_dev_male - std_dev_female)
49
50  def test_statistic(df):
51      range_per_county = df.groupby('county_idx')['y_rep'].apply(lambda x:
        x.max() - x.min())
52      average_range = range_per_county.mean()
```

## A.7 EXAMPLE QUALITATIVE FEEDBACK FROM DOMAIN EXPERTS

We collected feedback from statistics experts regarding the model criticisms produced by CALM. Here are two representative quotes:

> "I liked seeing code I could immediately run and check, it allowed me to take fast action to assess the situation."

> "I liked when I had code that immediately applied to the model."

## A.8 PROMPT USED FOR QUALITATIVE CRITERIA LLM JUDGES

---

**LLM Judge Prompt for Qualitative Criteria**

```
1 You are an expert in statistical modeling and data science. Your
      task is to determine which model criticism is more transparent.
2
3 Context:
4 Dataset Description: ...
5
6 Data Sample: ...
7
8 Column Description: ...
9
10 Model being criticized:
11 ```
12 ...
13 ```
14
15 Criticism A:
16 <Criticism A>
17 {criticism_a}
18 </Criticism A>
19
20 Criticism B:
21 <Criticism B>
22 {criticism_b}
23 </Criticism B>
24
25 Please evaluate the transparency of the criticisms based on the
      following criteria, assuming the intended evaluator is a data
      scientist or statistician:
26 - How clear is the methodology used to generate the criticism?
27 - How explicitly are the relevant parts of the dataset identified
      in the criticism?
28 - How unambiguous is the process of determining the criticism's
      conclusions?
29
30 First, provide a detailed analysis of how each criticism meets the
      criteria and compare Criticism A and Criticism B. Second, state
      "A" or "B" to indicate which criticism is more transparent.
31
32 Important:
33   - Avoid any position biases and ensure that the order in which
      the criticisms were presented does not influence your decision.
34   - Do not allow the length of the responses to influence your
      evaluation.
35   - Do not favor certain names of the criticisms.
36   - Be as objective as possible.
37
38 Provide your response in the following format:
39 Comparison: <Detailed reasoning and comparison to determine
      prefered criticism>
40 Final Response: <"A" or "B">
```

Figure 11: **LLM judge prompt for determining which model criticism is more transparent.** The same prompt structure, with corresponding judging criteria descriptions, is used for actionable and tailored judge prompts. The order of criticisms (CALM being Criticism A vs. Criticism B) is randomized to avoid position bias, and impartiality instructions are adapted from the RewardBench (Lambert et al., 2024) judge prompts.

## A.9    DETAILS ON EXPERIMENT 4

We ran the revision process for a single round, at a temperature of 0.0, using 3 proposals in total; we run multiple proposals because temperature 0.0 was not deterministic. We choose the top five test statistics $T_k$, ranked by p-value, where the Bonferonni-adjusted p-value $< 0.15$. We choose top five to limit the number of models fit since the fitting procedure is computationally-intensive. We choose the cutoff value by examining the spread of the Bonferonni-adjusted p-values; there were 18 significant discrepancies. We note that, while this threshold is larger than a typical p-value, discrepancies that do not meet the traditional significance levels may nevertheless be valuable in the context of a closed-loop model discovery process. A less stringent threshold allows us to evaluate lower-significance discrepancies that still may improve model performance.

