# OpenReview forum: "CALM: Critic Automation with Language Models"
_ICLR.cc/2025/Conference — ICLR 2025 Conference Withdrawn Submission_

### Official Review · Reviewer_Q2Ws · 2024-11-01

**Soundness:** 2
**Presentation:** 2
**Contribution:** 2
**Rating:** 3
**Confidence:** 4

**Summary:**

The paper describes a new system for providing criticism to scientific experiments that involve fitting a model to a given dataset. The current system supports settings of regression analysis, computational models, and Bayesian analysis. The authors construct testing scenarios for each settings and show how the criticism generated by CALM overcomes common challanges of naive LLM-based solutions.

**Strengths:**

S1: The idea is useful for scientific discovery, and generally the problem of experimental validation is important. Natural Language Criticism can also have educational value.

S2: The three settings of regression analysis, computational models, and Bayesian analysis are relevant and well chosen.

S3: The authors have also put some good thought into setting up the different scenarios (e.g. the spurious feature scenario was interesting) as illustrative case studies.

**Weaknesses:**

W1: My main concern with the current version is the validation through human studies. The human study is pretty small, just 3 PhD students and in addition to that, the outcomes of the study are not discussed in depth. Usually, for studies that are so small, one would want to see actual qualitative user feedback and insights, which are missing. For such qualitative studies, the common practice is also to report the background of the participants, how familiar they are with the data, the tests being used etc. Again, it is also hard to judge how significant the outcomes are here given the very small sample size. I would recommend the authors to work with a user researcher that can set up a study with at least 10-15 users and set up a rigorous qualitative evaluation. Appendix A7 has some notion of this but it is very limited.

W2: Section 3 can benefit from providing a clear architectural diagram that also describes how design decisions in CALM contribute to addressing common issues with Naive solutions. Currently, it is not clear when LLMs are used and how they interact with other parts of the system that are not LLM based.

W3: Some of the results are somewhat obvious. For example, the result on sliced statistics can also be considered as a general criticism that is quite known all around.

**Questions:**

Q1: Is it possible to automate part of the evaluation of CALM? For example, one can generate problem settings where certain data assumptions are encoded in generated data together with mistakes, and then the evaluation can check if CALM is able to identify these issues?

---

### Official Review · Reviewer_63Rb · 2024-11-04

**Soundness:** 2
**Presentation:** 1
**Contribution:** 2
**Rating:** 5
**Confidence:** 4

**Summary:**

**Summary**:
This paper proposes to use LLMs to automatically detect significant model-data discrepancies during scientific model discovery and to produce model critiques that can be used to guide model’s revision. Through synthetic and real-world experiments, the authors validate the ability of the proposed metric to correctly identify discrepancies in model-data. They also perform a small human study to validate the actionability, transparency, and specificity of the generated critiques.

**Methodology**:
-  The authors begin by identifying discrepancies between model predictions and data. To this end, they instruct gpt-4-turbo (the critic LLM) to generate test statistics functions that enable the detection of data-model discrepancies. The functions are generated based on the dataset metadata and a symbolic representation of the model.
- Then, using the model predictions and dataset samples, the authors run the proposed test statistics and compute the empirical value that they use to determine whether there exists a discrepancy at significance α (discrepancy).
- Given the test statistic function and the empirical p-values, the authors prompt gpt-4-turbo to generate a textual explanation that summarizes the discrepancy and its p-value (critique).

**Strengths:**

- Quantitative evaluation of the proposed methodology, including validation of the accuracy and error rate of the framework in a synthetic problem.
- Qualitative evaluation of the proposed methodology in terms of the generated critiques by both LLMs and humans. The qualitative evaluation concerned both the transparency and the usefulness of the criticism in helping an LLM-based system to revise models.

**Weaknesses:**

- **generalizability to other models**: the proposed framework is validated using gpt-4-turbo, a costly language model, which may compromise the applicability of the framework at scale. The paper could be further improved by showing how running the experiments using a cheaper model (e.g., gpt-4o) and/or open source models (e.g.,  Llama 3.1) would affect the obtained results.
-  **generalizability of the results**: the conducted experiments are either too simple (simple synthetic regression setting with 4 variables) or include few data-model pairs (6 in Section 4.2, 36 in Section 4.3, and 10 for the human studies), raising questions about the generalizability of the proposed framework to more complex datasets.
- **lack of meaningful baselines**: despite mentioning various model criticism techniques in Section 2, the authors limit their comparisons to simple naive baselines. For example, the authors could compare with a chain-of-thought prompting approach.
- few insights about the generated and correctness of summary statistics: while the authors provide one example in Section 4.3, the paper could be further improved by adding additional insights and contrasting the proposed discrepancies with commonly discussed discrepancies in the literature (e.g., do these resemble the ones commonly found by humans?)

**Questions:**

1. The proposed methodology consists of generating test statistic functions to be expressed in Python code based on the predictive model’s symbolic representation in STAN. How sensitive is the accuracy of your methodology (CALM) to the input format? In addition to adding a description about why the authors adopt an approach with different input-output formats, the authors could add an ablation experiment where they show the variability of the proposed approach (in terms of accuracy and hallucination) when representing the model using Python (instead of STAN).
2. It appears that the efficacy of the proposed methodology depends on the ability of the LLM to generate a diverse enough and correct set of summary statistics to validate. How diverse is the set of summary statistics? Did you experiment with different LLMs (other than gpt-4-turbo) to determine their efficacy as critics? To address this question, the authors should make the assumption more evident in the paper, as well as report the fraction of unique summary statistics generated by gpt-4-turbo and, if time allows, assess the ability of other models to generate different summary statistics.
3. A part of the methodology consists of generating the test statistic functions. However, the statistic functions shown in A.5 appear to be defined in terms of variables that are not defined, suggesting that in some cases the model is unable to generate correct test statistic functions. What is the frequency of incorrectly generated outputs for gpt-4-turbo?
4. What is the minimum number of samples necessary to consider the test statistic reliable?
5. It appears that the proposed methodology generates multiple test statistics that are executed simultaneously. However, this seems to be cost-inefficient as it is possible that the critic generates repeated information. Have you considered using an iterative process where the results of previous runs would guide future critics?
6. In line 292-294, the paper describes the naive LLM critic baseline. However, it appears that the naive LLM critic baseline and the propose methodology differ in the types of provided inputs. For instance, the symbolic representation of the predictive model is different from the one used for the proposed methodology (pymc program vs STAN). Is it possible that the observed behavioral differences are due to differences in the prompt and not due to the superiority of the proposed approach? To better support the claim that “Naive LLM critic consistently hallucinates but CALM does not” (line 263) the authors could conduct an additional experiment where the input format for the naive LLM and CALM are the same.
7. In Section 4.1 the authors evaluate the use of critiques in driving model revisions. The paper compares the number of proposed revisions across naive and proposed approach, but does not quantify whether the proposed revisions are _actually correct_ (i.e., lead to “better” models). The authors could add details about the percentage of _correct revisions_ that led to better models.
8. In Section 4.1 the generation hyperparameters were not specified. The authors could clarify what temperature was used to generate the critiques and revisions in Section 4.1. Moreover, in Section 4.2 the temperature=0.7 but in Section 4.3 the temperature=1 and temperature=0. These values could be better motivated.
9. In lines 364-365, the authors claim “CALM exhibits a favorable trade-off between the true positive rate and false positive rate, and significantly outperforms the baseline method”. However the analysis was conducted considering 6 model-dataset pairs, raising questions about the relevance of such analysis. How can we ensure that the observed results are not due to random chance?
10. The authors conduct a human study (using 10 examples and 3 annotators) in Section 4.3. How were the labels aggregated? What was the inter-annotator agreement in the task? The authors should also clarify wha having X% alignment preferences mean.

---

### Official Review · Reviewer_h4p3 · 2024-11-04

**Soundness:** 3
**Presentation:** 3
**Contribution:** 1
**Rating:** 3
**Confidence:** 3

**Summary:**

This paper introduces new framework that automates the process of model criticism in scientific research using LLM called CALM. Key things to this work is that the criticism is crucial for refining scientific models but model criticism require human experts to evaluate discrepancy between models and data. CALM addresses this problem by using LLMs to generate summary statistics and apply hypotheses testing to validate the significance of any discrepancies found in model.

 1. Automating Model Criticism, CALM automates the generation of critiques by identifying discrepancies between model predictions and data, ensuring they are statistically significant through hypothesis tests.
2. Avoiding Hallucinations, The framework tackles the issue of LLMs generating incorrect critiques by grounding its evaluations in real data and statistical
3. Improving Model Discovery, CALM significantly improves the performance of models when integrated into LLM-based model discovery systems, demonstrating its potential to accelerate scientific progress.

**Strengths:**

The paper makes an original contribution by addressing a gap in the use of large language models (LLMs) for scientific discovery: automated model criticism. While much research focuses on generating scientific models using LLMs this paper focuses on critiquing and improving models. Which is important to consider as it enhance scientific reliability on the models.

The paper addresses a critical aspect of scientific discovery by automating the critique process, which is essential for improving the reliability of scientific models. This has the potential to significantly advance scientific AI systems, increasing the models validitiy.

**Weaknesses:**

This paper is missing a clear baseline. The results may seem impressive but lack context without a baseline.
Motivation seems weak for this study. This paper should expand on why automating model criticism is necessary, emphasizing the limitations of traditional approaches or the risk when critiques are inaccurate.

 The paper does not fully explain how the LLM selects test statistics and tailors its critiques for different models. This is a critical aspect of the system but the process by which the LLM determines which statistics to use is only hinted at without further explanation.
Although the paper does focus on hypothesis testing to avoid hallucinations, there is limited discussion on how CALM handles false positives and false negatives in practice, especially in real-world scenarios. The results from synthetic datasets are promising, but further analysis on how it handles borderline cases in complex datasets. The paper does not discuss in detail what happens when the metadata is incomplete, messy, or unstructured.
The paper primarily focuses on certain types of models, such as regression and Bayesian models. These are important but lack of diversity in the models and datasets used for evaluation leaves questions about the generalizability of CALM.

**Questions:**

Can you provide more clarity on the specific logic or mechanisms used by the LLM to decide which test statistics are relevant for different models?
Why was the focus on regression and Bayesian models? Would you expect different results if you applied CALM to other model types?
How does CALM compare to existing tools for automated model evaluation?

---

### Official Review · Reviewer_h6mY · 2024-11-09

**Soundness:** 2
**Presentation:** 1
**Contribution:** 1
**Rating:** 3
**Confidence:** 2

**Summary:**

The authors propose CALM: a workflow that uses LLMs (GPT-4) to critique proposed statistical (scientific) generative models. CALM uses a prompt template to provide as input to an LLM information about a proposed statistical model, and prompts the LLM to propose test statistics that can surface "discrepancies" between the proposed statistical model, and true data-generating distribution. The authors also then prompt an LLM to generate a natural-language criticism of the proposed model, and also output code to "revise" the proposed model. In the paper, the authors evaluate their proposed workflow through a series of experiments where they use CALM to critique statistical models.

**Strengths:**

* The authors clearly motivate their context (developing scientific models of data) and argue convincingly why they believe automated criticism might help.
* The authors acknowledge several limitations of their automated approach, such as the possibility that CALM can hallucinate proposed code or natural language criticism.
* The authors design a series of different empirical evaluations to demonstrate the value of CALM. I do have several critiques of their evaluation methodology, which I enumerate below.

**Weaknesses:**

I have three primary critiques of this paper that influenced my recommendation to reject.

* **W1: Unclear novelty relative to related work, particularly Li et al. [1].** My primary concern is that the authors do not properly attribute, engage with, and distinguish what is novel in their proposed approach (CALM) from past work. In particular, the authors state that they adopt methods from Li et al. [1] as components of their proposed method (e.g., to generate code). When I read through this paper, I noticed that Li et al. _already_ propose a process to critique proposed scientific models, that seems _very similar_ to the authors' proposed process.
  * From Li et al: "in the criticism step, we ask the critic LM to produce natural language criticism of fitted models, we use this criticism to drive model revision. To enable the critic LLM to do something akin to a posterior predictive check, we obtain samples from the posterior predictive distribution and then compute summary statistics of these posterior predictive samples."
  * My understanding is that your methodology is very similar; but instead you have more flexibility for an LLM to propose exactly which summary statistics are computed?
  * The authors should update the draft to properly attribute Li et al.'s methodology and clearly distinguish how their methodology is different.

* **W2: Unclear notation and exposition makes it difficult to understand the author's proposed method.** Despite re-reading the main text and looking through the Appendices, I still found it difficult to understand the authors' proposed hypothesis testing method in Sections 3.1 and 3.2. I list points of confusion and make recommendations on how to better clarify your methodology below.
  * What exactly does it mean to "compute the test statistic over the model samples" to get the "null distribution"? For example, say that the test statistic is the average (dataset-level) difference in radon levels between the basement and first floor. From your notation, it looks like you will sample $m$ different _datapoints_, but it is unclear to me how this can create a distribution. Is the idea instead that you will sample $m$ different _datasets_, and then construct an empirical distribution of the value of the test statistic for those different _datasets_?
  * I am confused by the notation in Equation 2. I think what you're trying to say is that the empirical p-value is the proportion of test statistics (from the posterior samples) that are "more extreme" (greater than or equal to) the test statistic for the observed data.
  * I was trying to make the connection between your proposed methodology to calculate an empirical p-value and past literature on hypothesis testing using test statistics. If you're able to more explicitly distinguish what your methodology is inspired by/drawing from, and what exactly is distinct (that you're supporting customized test statistics), I'd find that helpful as a reader.

I've noted other places in the manuscript that are otherwise unclear and should be clarified by the authors:
  * I recommend that you make the draft clearer early on about assumptions that are necessary in order to use your proposed method. You frame the paper as broadly being able to aid with "scientific discovery/scientific modeling", but then it turns out that it seems like you're implicitly assuming the users' data columns are tabular and column names are "semantically plausible"; that sufficient "metadata" about the dataset is available (e.g., this approach may not be appropriate for image modeling/modeling bioinformatic data).
  * The title of the paper is not descriptive enough. I recommend that the authors revise the title to make clear that their contribution is on critiquing _scientific models_.
  * As early as possible, it may be helpful to give an example to illustrate clearly what you mean when you use the terms "discrepancy" (between what?), "summary statistic", "statistical test", "null distribution" (for what null hypothesis?). I only understood what you meant by these terms when I reached Section 4.1 where you contextualized each of these terms using the radon dataset.
  * I am still confused about what exactly is meant when you use the term "dataset metadata", and looking at Figure 9 did not help. Can you define precisely what you mean/what is given as input to the LLM? This is necessary information for a reader to know to use your proposed method.
* You should provide more details in your text about how exactly you used other systems. For example, exactly did you generate the model revisions from the critiques? All you do is cite that you used Li et al.'s "system", but more details should be included about how this system works. If Li's system does in fact support all of these things, why not just use it from the beginning to recommend the revisions itself (or as a more competitive baseline)?
  * For experiment 3, can you report the "win rate" for the human raters, rather than just the alignment between the human raters and LLM judges? Can you make clear if the win rates reported in Tables 1 and 2 are with LLM or human judges?


* **W3: The right baseline should be a _human_, not another LLM.** It seems to me like the LLM is relying a lot on "domain knowledge" about what features might be most important, based on their column names, to recommend test statistics. I wonder if a human would already hold this same intuition, and would come up with the same test statistics as CALM, without any LLM support. I would be interested if the LLM would result in a significant speed-up or would propose new test statistics that a human otherwise would not have thought of.

**Questions:**

See my questions embedded in the "Weaknesses" section.

---

### Note · Authors · 2024-11-28

I have read and agree with the venue's withdrawal policy on behalf of myself and my co-authors.